# Chemical Profiling, Bioactive Properties, and Anticancer and Antimicrobial Potential of *Juglans regia* L. Leaves

**DOI:** 10.3390/molecules28041989

**Published:** 2023-02-20

**Authors:** Natalia Żurek, Karolina Pycia, Agata Pawłowska, Leszek Potocki, Ireneusz Tomasz Kapusta

**Affiliations:** 1Department of Food Technology and Human Nutrition, College of Natural Sciences, University of Rzeszow, 4 Zelwerowicza St., 35-601 Rzeszow, Poland; 2Department of Biotechnology, College of Natural Sciences, University of Rzeszow, 1 Pigonia St., 35-310 Rzeszow, Poland

**Keywords:** *Juglans regia*, leaves, polyphenol compounds, antioxidant activity, anticancer activity, antibacterial activity, nutraceuticals

## Abstract

The aim of this study was to assess the biological potential of the polyphenolic fraction isolated from *J. regia* leaves, collected in the Subcarpathian region (Poland). The phenolic profile was determined using the UPLC-PDA-MS/MS method. Biological activity was determined by evaluating the antioxidant, anticancer, antibacterial, and antifungal effects. Prior to this study, the purified polyphenolic fraction was not been tested in this regard. A total of 40 phenolic compounds (104.28 mg/g dw) were identified, with quercetin 3-*O*-glucoside and quercetin pentosides dominating. The preparation was characterized by a high ability to chelate iron ions and capture O_2_^•−^ and OH^•^ radicals (reaching IC_50_ values of 388.61, 67.78 and 193.29 µg/mL, respectively). As for the anticancer activity, among the six tested cell lines, the preparation reduced the viability of the DLD-1, Caco-2, and MCF-7 lines the most, while in the antibacterial activity, among the seven tested strains, the highest susceptibility has been demonstrated against *K. pneumoniae*, *S. pyogenes,* and *S. aureus*. Depending on the needs, such a preparation can be widely used in the design of functional food and/or the cosmetics industry.

## 1. Introduction

Oxygen is essential for life. It is consumed in the process of aerobic respiration. As a result of metabolic changes, reactive oxygen species (ROS) are formed, which are responsible for the formation of inflammations, cellular disorders, cardiovascular diseases, vision disorders, or neurodegenerative diseases, such as Parkinson’s or Alzheimer’s disease. ROS can be, however, eliminated by antioxidants. Epidemiological studies have confirmed that the use of a diet based on vegetables and fruits rich in phenolic acids and flavonoids is important in the prevention of ROS-related diseases [1]. Among plant materials, walnut is a rich, natural source of antioxidants.

Walnut (*Juglans regia* L.) belongs to the Juglandaceae family. The leaders in the global production of walnuts are China, the United States, and Europe, but they are appreciated all over the world for their nutritional and sensory properties, as well as health benefits [2]. Walnuts are a rich source of polyphenols, the concentration of which depends on the date of harvest, variety [3,4,5], maturity level [6], geographical location [7], climatic and agrotechnical conditions. The edible parts of *Juglans regia* L. are kernels, containing in their composition polyunsaturated fatty acids (PUFA), vitamins, tocopherols, phytosterols, and several other bioactive substances important in the prevention of non-infectious diet-related diseases [3,4,5]. In addition, other parts of *Juglans regia* L., such as leaves, green shell (exocarp), woody shell (endocarp), flowers, pollen, and bark, are valuable products as well [2]. Walnut leaf infusion is used in traditional medicine in the treatment of venous insufficiency, as well as hemorrhoids, due to its antifungal, antibacterial, anthelmintic, astringent, and hypoglycemia properties [8]. Nour et al. (2013) indicate that these properties are related to the high content of tannins, phenolic acids, and flavonoids [9]. The global production of walnuts has also contributed to the formation of a huge amount of by-products that are incinerated, composted, or landfilled [2]. In turn, the woody shells serve as a source of energy for heating, an abrasive agents for cleaning and polishing metals, plastics, as a filter medium for separating oil, hazardous materials, and heavy metals [2,10]. Żurek et al. (2022 a,b) showed that both flowers and pollen from a walnut growing in the Subcarpathian region (Poland) have strong antioxidant, anti-microbial, and anti-carcinogenic potential and, thus, can be used as nutraceuticals or for food enrichment [11,12]. Additionally, Pycia et al. (2022) confirmed that walnut flowers may be an interesting functional additive to wheat bread, which showed an increased antioxidant potential [13].

The assessment of antioxidant and antimicrobial properties of by-products from the production of walnuts, such as leaves, is particularly important in terms of finding new, alternative sources of antioxidants and nutraceuticals, as well as their possible applications in the food or cosmetics industry.

The literature provides information on the antioxidant and antimicrobial potential of walnut leaves from trees growing in various regions of the world, but there is no comprehensive characterization of the polyphenol profile, antimicrobial, and anticarcinogenic properties of walnut leaves growing in the Subcarpathian region. Moreover, explanation of the mechanisms of action of such plant preparations will contribute to expanding the knowledge in this area.

## 2. Results

### 2.1. Total Phenolic, Flavonoid, and Proanthocyanidin Contents

Polyphenolic compounds, belonging to one of the most important groups of secondary metabolites, have been associated with several biological properties, such as antioxidant, anti-inflammatory, antibacterial, antiviral, and antiproliferative effects [10]. Their content in the preparation of *J. regia* leaves was assessed spectrophotometrically and expressed as total phenolic (TPC), flavonoid (TFC), and total proanthocyanidin contents (TPA). The results obtained are presented in Table 1. The TPC in the preparation obtained from *J. regia* leaves was 342.72 mg GAE/g dw, while the TFC and TPA were 55.64 mg QE/g dw and 26.24 mg CYE/g dw, respectively.

The estimated content of total proanthocyanidins and total flavonoids was significantly higher compared to previously published reports on the polyphenol analysis of *J. regia* leaves. This was mainly due to the fact that the obtained preparation was purified from ballast substances. In the work by Shah et al. (2018), the content of these two groups of compounds ranged from 37.61 to 46.47 mg GAE/g (TPC) and from 5.52 to 28.48 mg QE/g (TFC), respectively [14], and, in the work of Jabli et al. (2017), it was 103.33 mg GAE/g (TPC) and 20.17 mg QE/g (TFC), respectively [15], and, in the work of Giura et al. (2019), in the analysis of four varieties of *J. regia* leaves collected on six different dates, the content ranged from 25.10 to 104.25 mg GAE/g (TPC) and from 1.87 to 4.16 mg C/g (TFC) [16], while in the work of Untea et al. (2018), the TPC was 53.94 mg GAE/g [17]. In turn, the TPA content in the leaves of *J. regia* was assessed here for the first time, and only one report has been identified, which describes the content of TPA for other morphological parts of walnut [11]. In the cited study, the TPA content in the male flowers was estimated as about 1.6 times lower (16.66 mg CYE/g) compared to the results of our present analyses.

### 2.2. Antioxidant Activity

The growing interest in replacing synthetic antioxidants used as food additives with natural ones is conducive to research on plants to find new sources of natural antioxidants [10]. For these reasons, in this study, the antioxidant activity of *J. regia* leaves preparation was determined using five in vitro methods, such as scavenging activities (ABTS^•+^ radicals) (ABTS), scavenging reactive oxygen species (ROS) (hydroxyl radical (OH^−^), superoxide radical (O_2_^•−^)), and the ability of copper ion reduction (CUPRAC) and iron–ion chelation (ChA). The obtained results are presented in Table 2. The estimated superoxide, hydroxyl scavenging activity, and iron ion chelating capacity, expressed as IC_50_ values, were 67.78, 193.29, and 388.61 µg/mL, respectively. In turn, the antioxidant activity assessed by the ABTS^•+^ radical scavenging activity, and the ability to reduce copper ions was 9.09 and 1.16 mmol TE/g dw, respectively.

The presented results allow one to conclude that the preparation is characterized by high antioxidant activity, which is also confirmed by the data contained in other studies of *J. regia* leaves. In the work of Almeida et al. (2008), the estimated value for O_2_^•−^ radical scavenging was comparable (57.60 µg/mL) with the value obtained in this study, while the value for OH^−^ radical scavenging was about five times lower (1000 µg/mL) [18]. Equally high activity was found in the tested preparation concerning iron ion chelation. In the work of Erdogan Orhan et al. (2011), the estimated value for this parameter was about five times lower (>2000 µg/mL) [19]. Additionally, lower activity in other works was shown in relation to scavenging activities ABTS^•+^ radicals. Fernández-Agullo et al. (2019) and Untea et al. (2018) obtained values nearly seven times (1.26 mmol TE/g) [20] and 27 times lower (0.33 mmol TE/g), respectively [17]. In turn, the ability to copper ion reduction (CUPRAC) was assessed in the present study for the first time. Compared to the works evaluating the value of this parameter for other morphological parts of the *J. regia* tree, the estimated value for flowers was 2.9 times lower (3.33 mmol TE/g dw), while for pollen it was 3.6 times higher (0.32 mmol TE/g dw) than in this research [11,12].

The available literature data indicate that at least part of the observed antioxidant activity of plant preparations may be due to the presence of polyphenolic compounds [10,11,12]. Our analyses showed strong correlations between antioxidant activity and selected groups of polyphenolic compounds (TPC vs. CUPRAC, r > 0.999, *p* < 0.01; TFC vs. CUPRAC, r > 0.998, *p* < 0.05) (see Appendix A). The results demonstrate that the obtained preparation can be used as an easily available source of polyphenolic compounds, as well as a source of natural antioxidants. This justifies the use of the preparation in food design, as an auxiliary agent against oxidative damage, as well as a preventive and therapeutic agent in inflammatory diseases.

### 2.3. Cell Viability

Seven cell lines were selected to evaluate the effect of *J. regia* leaf preparation on cell viability. Six cancer lines, such as colon adenocarcinoma (DLD-1 and Caco-2), breast adenocarcinoma (MCF-7), glioblastoma (U87MG), astrocytoma (U251MG), melanoma (SK-Mel-29), and one line of healthy colon epithelial cells (CCD 841 CoN). The viability of the cells after 24, 48 and 72 h of treatment was assessed using the MTS test. The obtained results are presented in Figure 1 and Table 3.

The resulting preparation showed the highest cytotoxic activity against the DLD-1, Caco-2, and MCF-7 cell lines (see Figure 1). For the three cell lines mentioned, the estimated IC_50_ values were 214.11, 250.03 and 255.99 µg/mL, respectively, after 48 h treatment of the cells. The lowest cytotoxicity was assessed against malignant cell lines, such as U251MG and SK-Mel-29.

At the same time, the effect of the analyzed preparation on the viability of human colonic epithelial cells (CCD 841 CoN) was studied (see Figure 1). The IC_50_ values obtained ranged from 307.22 (72 h) to 501.10 (24 h) µg/mL. This result was lower compared to the cytotoxic effect assessed for the DLD-1, Caco-2 and MCF-7 cell lines, and it was comparable to the results reported for the other cancer cell lines.

Among the analyzed cell lines, only for the MCF-7 and Caco-2 lines, there are reports on cytotoxic activity of *J. regia* leaves. The anticancer activity of the preparation against the DLD-1, U87MG, U251MG, SK-Mel-29, and CCD 841 CoN cell lines was determined here for the first time. In the investigation of Carvalho et al. (2013), who analysed the cytotoxic activity of walnut leaf extracts against the Caco-2 cell line, the results are comparable (229.0 µg/mL) to those obtained in this study [21]. Similar results against the MCF-7 cell line were also received by Vieira et al. (2019) and Santos et al. (2013) (268.0 and 242.14 µg/mL, respectively) [22,23]. On the other hand, lower activity (ranging from 450.0 to 1500.0 µg/mL) in relation to the MCF-7 cell line was demonstrated in the work of Salimi et al. (2011) [24]. 

In conclusion, the strongest cytotoxic activity of the *J. regia* leaves preparation was found against benign tumor cell lines. Statistical analysis showed strong correlations between these cell lines and groups, as well as individual polyphenolic compounds (TPC vs. MCF-7, r > −0.999, *p* < 0.05; TFC vs. DLD-1, r > 0.998, *p* < 0.05) (see Appendix A). Such a relationship has already been reported, among others, in works on other morphological parts of the walnut tree, such as seeds [21], green husk [25], root bark [26], flowers [11], or pollen [12]. The presented results are extremely interesting in the context of the search for new sources of compounds effective in the prevention and treatment of cancer. In this sense, however, further research is needed to assess, among other things, the mechanism of cells death.

### 2.4. Antibacterial Potential

The antibacterial and antifungal activities of the *J. regia* leaves preparation were tested against three strains of Gram-negative bacteria, three strains of Gram-positive bacteria and one strain of fungus. The results are summarized in Figure 2. At three selected concentrations of aqueous walnut leaf extracts (10 mg/mL, 1 mg/mL; 0.1 mg/mL), only the concentration of 10 mg/mL showed strong bactericidal activity against the G (−) species *Klebsiella pneumoniae* > *Pseudomonas aeruginosa*. Less bactericidal activity was also observed against G (+) bacteria in the order *Staphylococcus aureus* > *Enterococcus faecalis* > *Streptococcus pyogenes*. In the case of *C. albicans*, the concentrations of aqueous extracts used did not show any antifungal effect.

Alcoholic extracts with a concentration of 10 mg/mL showed a bactericidal effect against all tested microorganisms in the following order: *Klebsiella pneumoniae* (bactericidal effect strongest) > *Staphylococcus aureus* > *Pseudomonas aeruginosa* > *Streptococcus pyogenes* > *Escherichia coli* > *Enterococcus faecalis*. Furthermore, the applied concentration of 10 mg/mL of alcoholic extract had a lethal effect on *Candida albicans*. At a concentration of 1 mg/mL, the antibacterial effect was maintained against G (−) *Klebsiella pneumoniae* and 2 species of G (+) bacteria: *Streptococcus pyogenes* > *Staphylococcus aureus*. For both extracts, the use of a concentration of 0.1 mg/mL did not show antimicrobial properties (Figure 2A,B).

Our results are consistent with other findings indicating that ethanol and water walnut leaf extracts are effective agents against many bacterial species [27,28,29]. Furthermore, in a study by Pereira et al. (2007), the effectiveness of aqueous extracts of the leaves of all *J. regia* varieties was demonstrated against mainly Gram-positive bacterial species, in the order of *B. cereus* > *S. aureus* > *B. subtilis*. The most sensitive microorganism for these studies was *B. cereus*, where aqueous extracts at a concentration of 0.1 mg/mL effectively inhibited the growth of this bacterium. Interestingly, in contrast to our results, Gram-negative species (*E. coli*, *P. aeruginosa,* and *K. peumoniae*), as well as yeasts (*C. albicans* and *C. neoformans*), were resistant to all water extracts from different walnut cultivars [8]. Our study shows the antibacterial efficacy of aqueous and alcoholic extracts, especially against Gram-negative *K. pneumoniae*. Aqueous extracts were used at 10 mg/mL, as well as alcoholic extracts at 10 mg/mL; 1 mg/mL effectively inhibited the growth of this bacterium (Figure 2A,B). Interestingly, other analyses indicate that Gram-negative bacteria are insensitive to plant extracts due to a different cell wall structure compared to Gram-positive bacteria [29,30]. Gram-negative bacteria are enveloped by a thin cell wall of peptidoglycan, which in turn is surrounded by an outer membrane containing lipopolysaccharide (LPS) and lipoproteins [31]. The presence of lipopolysaccharide (LPS) in the cell wall of Gram-negative bacteria has been shown to provide a hydrophilic environment, thereby protecting bacterial cells from hydrophobic molecules. This may explain why, in this study, aqueous and ethanol extracts, which may contain more hydrophilic compounds in their composition, showed greater inhibitory activity against *Klebsiella pneumoniae* or *Pseudomonas aeruginosa* [32].

It also appears that the discrepancies in the antimicrobial efficacy of the tested extracts may be due to the seasonal variability of antimicrobial active substances contained in the biomass of walnut leaves [33].

Similarly, it is indicated that extracts from various parts of walnut shows antimicrobial potential. Oliveira et al. (2008) examined aqueous extracts based on green husks of different varieties of walnut for antimicrobial properties against *B. cereus*, *B. subtilis*, *S. aureus*, *E. coli*, *P. aeruginosa*, *K. pneumoniae*, *C. albicans,* and *C. neoformans* [34]. The extracts showed antibacterial activity against G (+) bacteria in the series *S. aureus* > *B. cereus* > *B. subtilis*. The most sensitive was *S. aureus*, and its growth was already inhibited at a MIC concentration of 0.1 mg/mL. *S. aureus* is particularly dangerous due to the dangerous enterotoxins it produces, which can also be found in food. In a recent study, Żurek et al. (2022) showed that an aqueous–alcoholic extract of walnut flowers also has antimicrobial potential. According to the cited authors, the greatest antimicrobial activity could be observed against Gram-positive bacteria (*S. aureus* with a MIC 0.3125 mg/mL was the most sensitive). Gram-negative bacteria, on the other hand, were less sensitive, and the species *K. pneumoniae* was the most resistant with a MIC of 20 mg/mL) [35].

In addition, other experimental evidence has confirmed the antifungal effects of various extracts of *J. regia*, including leaves, bark, and roots [36,37,38,39]. However, with regard to *C. albicans*, the tested extracts at lower concentrations were less effective in limiting the growth of these pathogens. This effect may be due to the reduced permeability of the yeast cytoplasmic membrane to natural extracts and its greater ability to produce biofilms [40]. Furthermore, our analysis of the composition of natural phenolic compounds in leaf biomass revealed the presence of quercetin, which is often indicated along with myricetin and rutin as flavonoids with antifungal activity [39,41]. 

Furthermore, we speculate that the observed bactericidal effects at higher concentrations of the extracts may be due to the limited bioavailability of the active ingredients to the bacterial cells tested (CFU/mL) by the spot-on-lawn method. On solid media, cells grow as clusters (colonies), which may limit the penetration of bioactive compounds deep into the bacterial colony, in contrast to free-floating cells of the same bacterial strains in liquid cultures. A similar effect was observed with aqueous and alcoholic extracts of *Planktochlorella nurekis* at 100 mg/mL and 10 mg/mL using the spot-on-lawn method; however, the bactericidal effect in liquid culture was maintained at lower concentrations of the extracts, i.e., at 1 mg/mL, 100 μg/mL, and 10 μg/mL, respectively [42]. Taken together, these results demonstrate the potential of *J. regia* leaves preparation as an economical alternative source of antimicrobial agents due to the increasing problem of drug resistance observed among microorganisms.

### 2.5. Identification and Quantification of Phenolic Compounds

The chemical profile of the hydromethanolic extract of *J. regia* leaves was analyzed by UHPLC-DAD-ESI-MS. The chromatographic and spectral data of the detected compounds are shown in Table 4. The components were identified either by using reference, commercial standards (target identification) or by comparison of the retention times, elution orders, ESI-MS spectrometric data, and photodiode array PDA/UV-vis spectra with the literature data (tentative identification). In total, 40 compounds were identified, 14 of which were phenolic acid derivatives, four were tannins, 17 were flavonols, two were flavanonols, one was flavone, and two were naftoquinones.

Chlorogenic acid (compound **1**), caffeic acid (**8**), quercetin 3-*O*-xyloside (**16**), 3-coumaric acid (**17**), quercetin 3-*O*-glucoside (**19**), quercetin 3-*O*-galactoside (**20**), kaemferol 3-*O*-glucoside (**24**), quercetin 3-*O*-rhamnoside (**27**), and kaempferol 3-*O*-rhamnoside (**32**) were unambiguously recognized by comparison of their chromatographic and spectroscopic data with authentic standards.

Components **2**, **3**, **4**, **33**, and **40** were distinguished as caffeic acid derivatives by the characteristic UV spectra with two maximum peak absorption bands at 200 sh and 325–330 nm and typical fragmentation pattern with a daughter ion at *m*/*z* 179 [43]. Components **2** and **3** were identified as caffeoyl glucoside isomers; they produced the parent ion at *m*/*z* 341 [M-H]- and MS/MS ion at *m*/*z* 153 [M-H-162]-, indicating the loss of a hexose moiety. Their presence has been detected in *J. regia* leaves by several authors [22,44,45]. Constituents **4**, **33,** and **40** remain not fully characterized, however, they were already observed in our previous studies on the pollen and male flowers of walnut [11,12]. Peaks **5**–**7**, **9**, **11** were determined as coumarine derivatives. Components **5** and **6** were isomers of coumaroyl-quinic acid with different substituent positions. Both [M-H]- peaks at *m*/*z* 337 gave the same fragmentation pattern with the predominant ion at *m*/*z* 163 [M-H-174]-. Compounds **7**, **9**, and **11** gave [M-H]- at *m*/*z* 325, and their MS/MS spectrum revealed the presence of a high intensity fragment at *m*/*z* 163 [M-H-162]-, suggesting the loss of a hexose unit. Therefore, these were tentatively identified as coumaroyl glucosides. In fact, coumaric acid derivatives are commonly found in walnut leaves [8,9,22,23,33,44,45,46]. Pseudomolecular ion [M-H]- at *m*/*z* 355 (**12**) was assigned to ferulic acid hexoside, based on the MS/MS *m*/*z* at 193 [M-H-162]- and the literature findings [22,44].

Constituents **13–15** and **30** were identified as tannins. Tannins are polyphenolic compounds with a sugar core. Component **13** was identified as di-metoxycinnamoyl hexoside with a [M-H]- ion at *m*/*z* 369 and the characteristic fragment at *m*/*z* 207, indicating the loss of sugar unit. In turn, components were **14**, **15,** and **30** were characterized as gallotanins. Typical losses for gallotanins include gallic acid moieties (152 or 170 Da) and sugar units (162 or 180 Da) [47]. According to the proposed fragmentation pathway, compounds **14** and **15**, which exhibited a [M-H]- at *m*/*z* 467 and similar characteristic fragments at *m*/*z* 315, 169, 125, were identified as di-galloyl-deoxyhexoside isomers (**14**, **15**). Component **30** was identified as di-galloyl-shikimic acid. It gave a [M-H]- ion at *m*/*z* 477 and the fragment ions at 325, 169. These characteristic fragments are consistent with the literature [48].

Compound **10** was tentatively recognized as a naphtoquinone, dihydroxytetralone glucoside. Its LC-MS/MS spectrum consisted of a parent ion [M-H]- at *m*/*z* 339 and fragmentation ion at 159 [M-H-18-162]- showing the loss of a hexose residue and water, as previously reported by Vieira et al. (2019), Medic et al. (2021), and Gawlik-Dziki et al. (2014) [22,44,45]. To the same class of the components was assigned hydrojuglone derivative (**36**), which was recognized due to the distinctive daughter ion at *m*/*z* 175 and comparison to the available information regarding leaves of *J. regia* [44].

Compounds **22**, **23**, **25**, **26**, **28**, **29**, **31**, **34**, **35**, **37**, and **39** could be recognized as flavones by their characteristic UV-Vis absorption spectra. The methanol spectra of flavones and flavonols exhibit two major absorption peaks in the region 240–400 nm [49]. Further, compounds **22**, **23**, **25**, **34**, **35,** and **37** all gave the deprotonated aglycone fragment at *m*/*z* 301, suggesting that they were originating from quercetin. Components **22**, **23**, and **25** were recognized as quercetin pentosides. These all showed the loss of 132 u. It is difficult, however, to identify the isomers of sugar by mass spectrometry alone [50]. Constituent **35** yielded [M-H]-42, u, which is higher than compounds **22**, **23**, and **25**, indicating the occurrence of acetyl moiety in the molecule and thus, being identified as quercetin acetyl-pentoside. Similarly, compounds **34** and **37** were identified as acetyl derivatives of quercetin. In this case, however, a loss of 188 u [M-H-42-146]- was observed, indicating the presence of rhamnose unit. Indeed, the presence of acetyl derivatives of quercetin in the leaves of *J. regia* have already been defined [22,45]. Compounds **26**, **28**, **29**, **31**, and **39** instead produced a deprotonated aglycone fragment at *m*/*z* 285, which indicated that they were originating from luteolin or kaempferol. However, their UV spectra with absorption maxima at ca. 260 and 340 nm were characteristic for kaempferol [51]. Moreover, kaempferol and its glycosylated derivatives are commonly found in the walnut leaves [8,9,22,23,33,44,45,46,52]. This led to the aglycone identification as kaempferol. Further, constituent **26** had [M-H]- at *m*/*z* 447. The MS/MS spectrum showed the loss of 162 u and, thus, the compound being kaempferol hexoside. Compounds **28**, **29**, and **31** gave the same deprotonated ion at *m*/*z* at 417 and the same fragmentation spectra at *m*/*z* 285 [M-H-132]-. The loss of 132 u indicated the loss of a pentose moiety. These components were identified as kaempferol pentosides. Component **39** with the parent ion at *m*/*z* 473 and the fragmentation pattern [M-H-42-146]- was found to be kaempferol acetyl-rhamnoside. This is in line with the previous investigations [45].

Constituents **18** and **21** were distinguished as flavanonols. Both gave [M-H]- at *m*/*z* 435 with identical daughter ions at *m*/*z* 285. This fragmentation pattern is characteristic for taxifolin pentoside isomers and was observed by Santos et al. (2013) in the studies on the walnut leaves found in the Portuguese samples [23].

Finally, compound **38** was potentially identified as a biflavonoid, 4′′′-dehydroxyamentoflavone, based on the [M-H]- ion at *m*/*z* 521, and the daughter ion at *m*/*z* 375 in the MS/MS spectrum as reported by Yao et al. (2017) [53]. This constituent has been identified in *J. regia* for the first time.

Results obtained from quantitative analyses are shown in Table 4. The total amount of the identified secondary metabolites was 104.28 mg/g of dried matter. Quercetin 3-*O*-glucoside was found to be the most abundant compound (18.58%), followed by quercetin pentoside isomers (11.7% and 6.53%) and 3-*O*-coumaroylquinic acid (5.26%). Our analyses are in good agreement with already existing data. Additionally, other authors have reported, as major constituents of walnut leaves, coumaroylquinic acids and quercetine derivatives [8,22,23,46,52,54,55]. However, their quantities are difficult to compare due to variation of solvent and extraction method used. Moreover, the discrepancies in the chemical composition may also depend on species, growing conditions, and genetic differences [56].

## 3. Materials and Methods

### 3.1. Materials and Reagents

Quercetin, gallic acid, cyanidin chloride, neocuproine, ferrozine, NBT (nitrotetrazolium blue chloride), PMS (phenazine methosulfate), NADH (*β*-Nicotinamide adenine dinucleotide, reduced disodium salt hydrate), 2-Deoxy-D-ribose, and EDTA (ethylenediaminetetraacetic acid disodium salt dihydrate) were used. Dulbecco’s Modified Eagle Medium-GlutaMAX-1 (DMEM), RPMI 1640 Media, 0.25% trypsin-EDTA, fetal bovine serum (FBS), phosphate-buffered saline (PBS), antibiotics (100 U/mL penicillin, and streptomycins) were purchased from Sigma-Aldrich (Steinheim, Germany). Chlorogenic acid, caffeic acid, quercetin 3-*O*-galactoside, kaempferol 3-*O*-glucoside, quercetin 3-*O*-rhamnoside, quercetin 3-*O*-xyloside, quercetin 3-*O*-glucoside, and kaempferol 3-*O*-rhamnoside were obtained from Extrasynthese (Lyon, France). CellTiter 96^®^ AQueous Non-Radioactive Cell Proliferation Assay was purchased from Promega (Madison, WI, USA). All other chemicals were purchased from Chempur (Piekary Śląskie, Poland).

### 3.2. Plant Material

The leaves were collected from the *J. regia* tree in the Subcarpathian region in Poland in May 2020. Then, the material was frozen, dried in a lyophilizer (ALPHA 1–2 LD plus) (Osterode, Germany), and stored at −20 °C.

### 3.3. Preparation of Extract

The *J. regia* leaf preparation was obtained using the solid phase extraction method in accordance with our previous reports [35]. Briefly, the ground material (5 g) was mixed with methanol (50%; 45 mL) and sonicated in ultrasonic bath for 30 min at 30 °C (Sonic 10, Polsonic, Poland). The suspension was centrifuged at 19,000× *g* for 10 min (Centrifuge 5430, Eppendorf, Hamburg, Germany), the supernatant was collected, and the residue resubmitted to the above extraction using methanol (70%; 45 mL). The resulting supernatants were combined and concentrated using a rotary evaporator at 40 °C (R-215 Rotavapor System, Buchi, Switzerland). Concentrated samples were applied to the SPE column (LiChroprep RP-18; pore size 40–63 µm), previously conditioned with methanol, and equilibrated with water. The polyphenol fraction was eluted with methanol. The methanolic polyphenol extract was evaporated in a vacuum, lyophilized, and the obtained preparation was used for analysis.

### 3.4. Determination of Total Phenolic, Flavonoid and Proanthocyanidin Content

The total phenolic content (TPC) was evaluated using the method described by Gao et al. (2000) [57]. The plant extract was mixed with distilled water (2.0 mL), Folin-Ciocalteau reagent (0.2 mL), and 20% sodium carbonate (1.0 mL). After 1 h, the absorbance was measured at a wavelength of 765 nm using a UV–VIS spectrometer (Type UV2900, Hitachi, Tokyo, Japan).

The total flavonoid content (TFC) was estimated by following the procedure developed by Chang et al. (2020) [58]. The extract was mixed with ethanol (1.5 mL), aluminum chloride (0.1 mL), distilled water (2.8 mL), and 1 M sodium acetate (0.1 mL). After 30 min, the absorbance was measured at 415 nm. 

The total proanthocyanidin content (TPA) was determined according to the method described by Żurek et al. (2022) [12]. The extract was mixed with n-BuOH in 35% HCl (3.0 mL) and 2% iron (III) ammonium sulfate (0.1 mL). After incubation at 95 °C for 45 min, the absorbance was measured at 550 nm. 

The results of TPC, TFC, and TPA contents were expressed in milligram equivalent of gallic acid per gram of dry weight (mg GAE/g dw), milligram equivalent of quercetin per gram of dry weight (mg QE/g dw), and milligram equivalent of cyanidin chloride per gram of dry weight (mg CYE/g dw), respectively.

### 3.5. Determination of Antioxidant Activity

#### 3.5.1. Superoxide Radical Scavenging Activity Assay (O_2_^●−^ Method)

Superoxide radical scavenging activity was measured based on the method described by Robak and Gryglewski (1988) [59]. The plant extract was mixed with 150 µM NBT (1.0 mL), 468 µM NADH (1.0 mL), and 60 µM PMS (1.0 mL). After 5 min, the absorbance was measured at a wavelength of 560 nm. 

#### 3.5.2. Hydroxyl Radical Scavenging Activity Assay (OH^●^ Method)

Hydroxyl radical scavenging activity was evaluated by the method of Żurek et al. (2022) [12]. The plant extract was mixed with 0.2 mM 2-deoxyribose, 1.0 mM iron ammonium sulphate, 1.04 mM EDTA, 1.0 mM ascorbic acid, 0.1 M perhydrol, 2.8% trichloroacetic acid, and 1% thiobarbituric acid. After being heated to 100 °C for 15 min and cooled to room temperature, the absorbance was measured at 532 nm.

#### 3.5.3. Chelating Ability of Ferrous Ion (ChA Method)

The chelating ability of ferrous ions was assessed according to the method described by Żurek et al. (2022) [12]. The plant extract was mixed with 0.1 mM iron II sulfate (0.2 mL) and 0.25 mM ferrozine (0.4 mL). After 10 min, the absorbance was measured at 562 nm.

#### 3.5.4. ABTS^●+^ Radical Scavenging Activity (ABTS Method)

The scavenging activity of leaves extracts on ABTS^●+^ radicals was determined according to the method of Re et al. (1999) [60]. The plant extracts were mixed with ABTS^●+^ solution (0.03 mL), and then they were diluted with distillated water to an absorbance of 0.7. After 6 min, the absorbance was measured at 734 nm.

#### 3.5.5. Determination of Copper Ion Reduction (CUPRAC Method)

The CUPRAC test was determined by a spectrophotometric method described by Apak et al. (2006) [61]. The plant extract was mixed with 10 mM copper chloride (1.0 mL), 7.5 mM neocuproine (1.0 mL), and 1 M acetate buffer (1.0 mL). After 30 min, the absorbance was measured at 450 nm.

The results of scavenging activities superoxide radicals (O_2_^●−^), hydroxyl radicals (OH^−^), and the ability to iron ion chelation (ChA) were expressed as the values of inhibition concentration (IC_50_). The results of scavenging activities of ABTS^●+^ radicals (ABTS) and copper ion reduction (CUPRAC) were expressed as Trolox Equivalent per g of dry weight (mmol TE/g dw).

### 3.6. Cell Culture 

In the analysis, human cell lines CCD 841 CoN (human colon epithelial cells), Caco-2, DLD-1 (colon adenocarcinoma), MCF-7 (breast adenocarcinoma), SK-Mel-29 (melanoma), U87MG (glioblastoma), and U251MG (astrocytoma) were used. Cell lines were cultured in DMEM or RPMI 1640 medium supplemented with 10% FBS, 4 mM L-glutamine, 20 mM sodium bicarbonate, 5000 IU penicillin, and 5000 mg/mL streptomycin. Cells were incubated at 37 °C in a humidified atmosphere under 5% of CO_2_ (CB170 incubator, Binder, Tuttlinen, Germany).

### 3.7. MTS Cell Viability Assay

Cell viability was assessed using the MTS test, according to our previous reports [35]. Cell lines were seeded (8 × 10^3^ cells/well) in 96-well microplates and incubated in a humidified atmosphere (37 °C; 24 h) to promote adhesion. Then, cells were treated in triplicate at concentrations of 10, 100, 250, 500, 750 µg/mL *J. regia* leaves extract. The extracts were dissolved in water and then diluted in culture medium. After 24, 48, and 72 h of incubation, the MTS assay was performed according to the manufacturer’s protocol (Promega, Madison, WI, USA) using a microplate reader (SmartReader 96, Accuris Instruments, Edison, NJ, USA). The results were expressed as the IC_50_.

### 3.8. Antimicrobial and Antifungal Activity

The antimicrobial potency of *J. regia* leaves preparation has been determined using the following microorganisms: three strains of Gram-negative (*Escherichia coli* PCM 2209, *Klebsiella pneumoniae* DSM 30104, *Pseudomonas aeruginosa* DSM 19880) and three strains of Gram-positive (*Staphylococcus aureus* DSM 104437, *Streptococcus pyogenes* PCM 2318, and *Enterococcus faecalis*).

Additionally, the antifungal activity of the tested extracts was checked using *Candida albicans* ATCC 14,053 strain. Clinical strains of *Enterococcus faecalis* were received from the collections of microorganisms of the Frederic Chopin Provincial Specialist Hospital in Rzeszow, Poland, which were obtained during routine diagnostic cultures.

The microorganisms were cultured on Nutrient Agar NA (meat extract—10 g/L, peptone—10 g/L, sodium chloride—5 g/L, agar—20 g/L, pH—7.0) and YPD medium (yeast extract—10 g/L, peptone—20 g/L, glucose—20 g/L, pH—7.2). The water and ethanolic extracts were used to prepare dilutions, i.e., 1:10, 1:100, with which the tested microorganism species were treated. Antibacterial and growth inhibitory properties of *J. regia* leaves extract against selected microorganisms were tested using a spot-on-lawn method. Briefly, microorganism assay plates were prepared whereby the tested bacteria and yeast (8-log CFU/mL) were seeded. Then 5 µL of each *J. regia* leaves extracts was surface spotted onto the indicator lawn. In the case of ethanol extracts, positive controls of appropriately diluted ethanol, i.e., 80%, 8%, 0.8%, were also surfaced spotted onto the indicator lawn. Plates were allowed to incubate overnight. Growth inhibition was evaluated after incubation (24 h at 37 °C—bacteria and 29 °C—yeast) by observations of the zone of inhibition around the spots with the tested microorganism and were recorded using a Canon EOS 600D camera.

### 3.9. Determination of Polyphenols Profile by UPLC-Q-TOF-MS

Polyphenolic compounds were identified and quantified using the Ultra-Performance Liquid Chromatography Array Detector (UPLC-Q-TOF-MS, Waters, Milford, MA, USA) according to the protocol described by Żurek et al. (2021) [35]. Briefly, the separation of individual phenols was performed at 50 °C, using a UPLC BEH C18 column (100 mm × 2.1 mm, 1.7 µm, Waters, Warsaw, Poland), with an injection volume of 5 µL and an isocratic rate flow rate 0.35 mL/min. Solvent A (water) and solvent B (40% acetonitrile in water, *v*/*v*) were used as the mobile phase. The following parameters were used for triple-quadrupole detection: gas flow con 100 L/h; voltage 30 V; capillary voltage 3.5 kV; source temperature 120 °C; desolvation temperature 350 °C and desolvation gas flow 800 L/h. Results are expressed in mg/g dw.

### 3.10. Statistical Analysis

All analyzes were performed in triplicate and are presented as mean ± SD. Duncan’s, Tukey’s HSD, Student’s *t*-test (*p* < 0.05; *p* < 0.01; *p* < 0.001), and Pearson’s correlation (*p* < 0.05; *p* < 0.01) were analyzed using Statistica 13.3 (StatSoft, Krakow, Poland).

## 4. Conclusions

Walnut leaves are one example of the by-products associated with the production of the nuts. They have long been used in folk medicine as an effective agent with anti-inflammatory, antidiarrheal, anthelmintic, antiseptic, and astringent properties. The obtained research results indicate and confirm that they are a rich, valuable source of bioactive substances with a strong pro-health potential. Tests related to the analysis of antimicrobial activity indicate the high effectiveness of the alcohol extract of walnut leaves against the G (−) *Klebsiella pneumoniae* bacteria, i.e., a deadly bacteria responsible for inflammation of the respiratory, digestive, urinary systems, or meningitis. The extract from the tested walnut leaves was also characterized by very good activity against the G (+) *Staphylococcus aureus* bacteria, i.e., a dangerous human pathogen with high pathogenicity and mortality. Moreover, strong cytotoxicity of the tested extract was demonstrated towards cancer cells of colon adenocarcinoma (DLD-1 and Caco-2), breast adenocarcinoma (MCF-7), and cancers common in the human population. At the same time, out of seven analyzed cell lines, the cytotoxicity of this raw material was assessed for the first time for five. The described research results are a comprehensive statement confirming the strong health-promoting potential of the walnut leaf extract. On this basis, application possibilities of walnut leaves as nutraceuticals, enriching food or as components of cosmetics, should be sought.

## Figures and Tables

**Figure 1 molecules-28-01989-f001:**
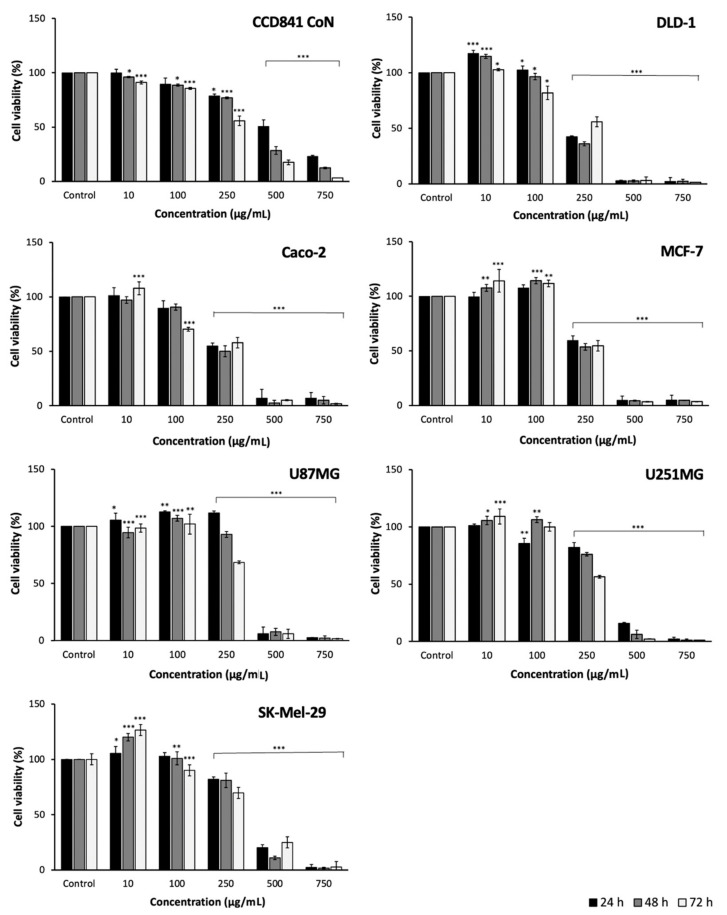
Effect of preparation of *J. regia* leaves on the cell viability of CCD 841 CoN, DLD-1, Caco-2, MCF-7, U87MG, U251MG, and SK-Mel-29. The cells were treated extracts in five concentrations (10–750 µg/mL) for 24, 48 and 72 h. The number of viable control (non-treated) cells at each time point served as 100%. Graphs represent mean values ± SD. Asterisks indicate a statistically significant differences (* *p* < 0.05, ** *p* < 0.01, *** *p* < 0.001) among samples according to Student’s *t*-test.

**Figure 2 molecules-28-01989-f002:**
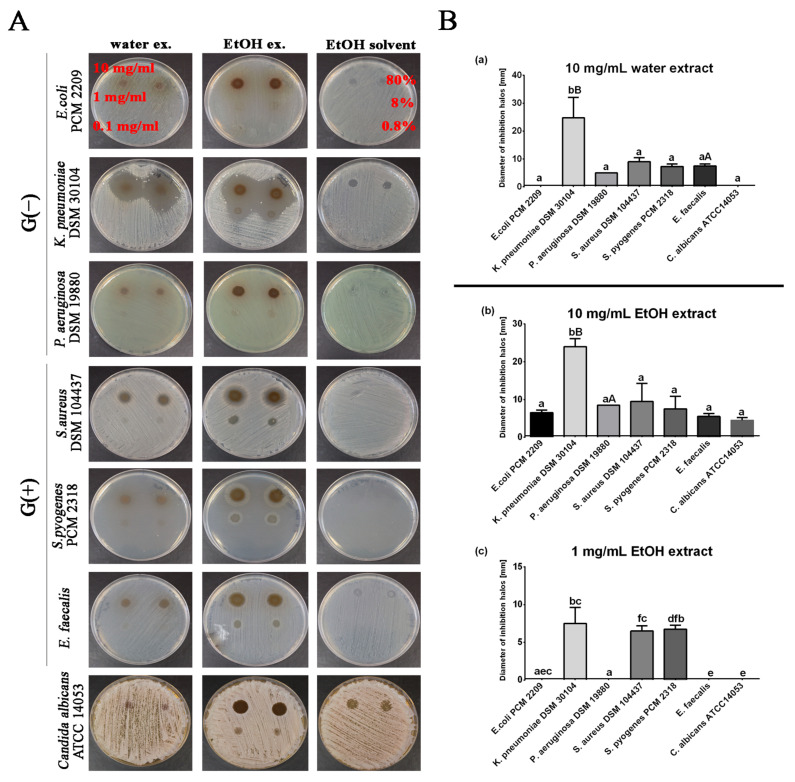
(**A**) Water and ethanolic *J. regia* extracts-mediated changes on microbial growth: G (−) *Escherichia coli* PCM2209, *Klebsiella pneumoniae* DSM 30104, *Pseudomonas aeruginosa* DSM 19880; G (+) *Staphylococcus aureus* DSM 104437, *Streptococcus pyogenes* PCM 2318, and *Enterococcus faecalis*; *Candida albicans* ATCC14053. The tested microorganisms were treated with *J. regia* leaves extracts at 10 mg/mL, 1 mg/mL, 0.1 mg/mL surface spotted onto the indicator lawn—NA agar (bacteria) and YPD agar (*Candida*) media. Representative micrographs of bacterial and yeast culture dishes; (**B**) diameter of growth inhibition (halo) of bacteria induced by walnut extracts; (**a**) 10 mg/mL water extract; (**b**) 10 mg/mL alcoholic extract; (**c**) 1 mg/mL alcoholic extract. The values are expressed as means ± SD. Letters indicate a statistically significant differences (capital letter < 0.01, small letter < 0.001) among samples according to ANOVA and HSD Tukey’s.

**Table 1 molecules-28-01989-t001:** The contents of total phenolics (TPC), flavonoids (TFC), and proanthocyanidins (TPA) in the preparation of *J. regia* leaves.

	TPC	TFC	TPA
	(mg GAE/g dw)	(mg QE/g dw)	(mg CYE/g dw)
*J. regia* leaves	342.72 ± 0.49	55.64 ± 0.06	26.24 ± 0.01

Abbreviations: GAE, equivalent of gallic acid; QE, equivalent of quercetin; CYE, equivalent of cyanidin chloride. Values are expressed as mean ± SD.

**Table 2 molecules-28-01989-t002:** Scavenging activities ABTS^•+^ radicals (ABTS), superoxide radicals (O_2_^•−^), hydroxyl radicals (OH^−^), and the ability to copper ion reduction (CUPRAC) and iron ion chelation (ChA) of the *J. regia* leaves preparation.

	O_2_^•−^	OH^−^	ChA	ABTS	CUPRAC
	IC_50_ (µg/mL)			(mmol TE/g dw)	
*J. regia* leaves	67.78 ± 0.94	193.29 ± 2.80	388.61 ± 1.62	9.09 ± 0.09	1.16 ± 0.01

Abbreviations: TE, trolox equivalent. Values are expressed as mean ± SD.

**Table 3 molecules-28-01989-t003:** IC_50_ values of *J. regia* leaves preparation against of seven cell lines after treatment for 24, 48, and 72 h.

No.	Cell Line	*J. regia* Leaves		
		IC_50_ (µg/mL)		
		24 h	48 h	72 h
1	CCD 841 CoN	501.10 ± 4.16	388.58 ± 10.05	307.22 ± 2.22
2	DLD-1	244.18 ± 4.27	214.11 ± 4.59	267.34 ± 19.64
3	Caco-2	270.65 ± 22.65	250.03 ± 10.27	276.02 ± 11.97
4	MCF-7	307.03 ± 1.90	255.99 ± 13.81	282.10 ± 5.71
5	U87MG	377.33 ±13.84	370.87 ± 13.20	327.11 ± 0.63
6	U251MG	379.05 ± 16.78	340.90 ± 13.82	285.24 ± 5.33
7	SK-Mel-29	379.89 ± 15.80	360.46 ± 3.04	361.55 ± 2.21

Abbreviations: CCD 841 CoN, colon epithelial cells; DLD-1, and Caco-2 colorectal adenocarcinoma cells; MCF-7, breast adenocarcinoma cells; U87MG, glioblastoma cells; U251MG, astrocytoma cells; SK-Mel-29, melanoma cells. Values are expressed as mean ± SD.

**Table 4 molecules-28-01989-t004:** Individual phenolic compounds identified by UPLC-PDA-MS/MS in *J. regia* leaves preparation.

No.	Compound	Rt	λ_max_	[M-H] *m*/*z*		Content
		min	nm	MS	MS/MS	mg/g dw
1	Chlorogenic acid *	2.27	299 sh, 327	353	191	2.78 ± 0.01 ^jk^
2	Caffeic acid 3-*O*-glucoside	2.48	299 sh, 324	341	179	1.30 ± 0.00 ^g^
3	Caffeic acid 4-*O*-glucoside	2.62	299 sh, 324	341	179	0.42 ± 0.02 ^ab^
4	Undefined caffeic acid derivative	2.71	279	463	179	0.43 ± 0.08 ^ab^
5	3-*O*-Coumaroylquinic acid	2.79	310	337	163, 119	5.49 ± 0.08 ^p^
6	4-*O*-Coumaroylquinic acid	2.94	310	337	163, 119	1.27 ± 0.06 ^g^
7	3-*O*-Coumaric acid glucoside	3.06	312	325	163	3.76 ± 0.11 ^m^
8	Caffeic acid *	3.23	299 sh, 324	179	135	1.98 ± 0.11 ^h^
9	4-*O*-Coumaric acid glucoside	3.36	312	325	163	0.86 ± 0.01 ^ef^
10	4,8-dihydroxy-tetralone-4-*O*-glucoside	3.43	310	339	159	1.27 ± 0.01 ^g^
11	4-*O*-Coumaric acid glucoside	3.48	312	325	163	0.73 ± 0.01 ^def^
12	Ferulic acid 4-*O*-glucoside	3.55	320	355	193, 175	0.79 ± 0.03 ^def^
13	Di-metoxycinnamoyl hexoside	3.65	328	369	207, 189	0.42 ± 0.05 ^ab^
14	Di-galloyl-deoxyhexoside isomer I	3.73	253	467	315, 169	0.37 ± 0.03 ^ab^
15	Di-galloyl-deoxyhexoside isomer II	3.82	253	467	315, 125	0.44 ± 0.03 ^bc^
16	Quercetin 3-*O*-xyloside *	3.96	255, 354	433	301	0.69 ± 0.02 ^de^
17	3-Coumaric acid	4.13	310	163	119	2.53 ± 0.10 ^ij^
18	Taxifolin pentoside isomer I	4.27	290	435	285, 151	0.63 ± 0.14 ^cd^
19	Quercetin 3-*O*-glucoside *	4.35	255, 353	463	301	19.38 ± 0.50 ^s^
20	Quercetin 3-*O*-galactoside *	4.35	255, 353	463	301	4.76 ± 0.22 ^o^
21	Taxifolin pentoside isomer II	4.89	290	435	285, 151	1.33 ± 0.10 ^g^
22	Quercetin pentoside isomer I	4.95	255, 354	433	301	2.11 ± 0.19 ^ij^
23	Quercetin pentoside isomer II	5.06	255, 354	433	301	6.81 ± 0.29 ^q^
24	Kaempferol 3-*O*-glucoside *	5.08	264, 338	447	285	3.03 ± 0.04 ^k^
25	Quercetin pentoside isomer III	5.15	255, 354	433	301	12.20 ± 0.42 ^r^
26	Kaempferol hexoside	5.34	264, 338	447	285	0.66 ± 0.02 ^de^
27	Quercetin 3-*O*-rhamnoside *	5.36	255, 354	446	301	4.36 ± 0.02 ^n^
28	Kaempferol pentoside isomer I	5.51	264, 338	417	285	2.30 ± 0.05 ^hi^
29	Kaempferol pentoside isomer II	5.60	264, 338	417	285	0.34 ± 0.05 ^ab^
30	Di-galloyl-shikimic acid	5.74	290	477	325, 169	1.99 ± 0.12 ^gh^
31	Kaempferol pentoside isomer III	5.51	264, 338	417	285	3.23 ± 0.27 ^l^
32	Kaempferol 3-*O*-rhamnoside *	6.01	264, 338	431	285	0.56 ± 0.01 ^bc^
33	Unidentified caffeic derivative	6.24	299 sh, 321	501	179	0.90 ± 0.04 ^ef^
34	Quercetin acetyl-rhamnoside isomer I	6.37	255, 354	489	447, 301	3.76 ± 0.00 ^m^
35	Quercetin acetyl-pentoside	6.38	255, 354	475	433, 301	2.12 ± 0.05 ^ghi^
36	Hydrojuglone derivative	6.68	327	517	175	0.30 ± 0.01 ^a^
37	Quercetin acetyl-rhamnoside isomer II	6.80	255, 354	489	447, 301	0.83 ± 0.04 ^ef^
38	4′′′-Dehydroxyamentoflavone	6.95	266, 312	521	375	4.30 ± 0.14 ^n^
39	Kaempferol acetyl-rhamnoside	7.12	264, 336	473	431, 285	0.84 ± 0.03 ^ef^
40	Unidentified caffeic derivative	7.89	299 sh, 324	501	179	2.01 ± 0.13 ^gh^
	Total					104.28 ± 2.57

Abbreviations: Rt, retention time; UV-Vis, ultraviolet-visible; [M-H]^−^, negative ion values; *m*/*z*, mass-to-charge ratio; *, compounds identified by standards. Values are expressed as mean ± SD. The statistical significance (values marked with different letters, a–s) was analyzed with Duncan’s test (*p* < 0.05).

## Data Availability

Data are contained within the article and Appendix A.

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
