# Peer review of "Chemical Profiling, Bioactive Properties, and Anticancer and Antimicrobial Potential of Juglans regia L. Leaves"

_molecules, 2023, doi:10.3390/molecules28041989_

Round 1

Reviewer 1 Report

The article is written extremely professionally, in a very good scientific style and language.

The experimental design is structured adequately to the set objective. Modern adequate methods for the analysis of the biological activities of Juglans regia L. leaves have been selected, and the results are described and commented on with great precision. This shows the excellent awareness of the authors in the field of oxidative stress, antioxidant activity, and redox-modulating properties of plant extracts. The scientific literature cited is appropriate and well-founded.

I congratulate the team for a precise and excellent job and propose that the article be accepted for publication in the journal Molecules.

Author Response

The authors thank you very much for such a favorable review. We are very grateful for the proof of recognition and emphasizing our professionalism. For a scientist, these are very important words that show commitment to the development of a scientific discipline.

Reviewer 2 Report

The manuscript (2195462) entitled “Chemical profiling, bioactive properties, anticancer and antimi-2 crobial potential of Juglans regia L. leaves” is significant to publish in Molecules. I recommend this manuscript for publication.

Author Response

The authors are very grateful for the positive review. We perceive the lack of critical comments as recognition of the scientific level we have achieved in conducting scientific research

Reviewer 3 Report

In the manuscript entitled “Chemical profiling, bioactive properties, anticancer and antimicrobial potential of Juglans regia L. leaves” by Natalia Å»urek et al., the authors study the chemical composition and biological potential of Juglans regia leaves.  They evaluated in vitro antioxidant activity, antimicrobial activity against 7 different strains, and cytotoxic activity against 6 different cancer cell lines of methanolic polyphenolic extracts of the leaves. In addition, they determined the polyphenolic profile of the extracts by using a UHPLC-DAD-ESI-MS approach.

The results shown in the manuscript confirm that walnut leaves, thanks to their high levels of bioactive substances have a high pro-health potential, thus suggesting possible applications either as nutraceuticals or as components of cosmetics.

Overall, the manuscript is suitable for the journal scope, is quite well written and the results are clearly reported. The part of the work concerning the determination of the polyphenolic content of the extracts and the subsequent identification and quantification of the phenolic compounds is accurate and constitutes the most robust part of the results. The part concerning the determination of the biological activity is instead a little weaker. The authors hypothesize an anti-tumor potential of walnut leaf extracts based on a non-specific antioxidant and cytotoxic/antiproliferative activity but do not provide indications on the molecular mechanisms of these activities, which however they declare to be the subject of future work.

Despite these objective limitations, the manuscript can be improved and could be suitable for publication after the authors have responded to the following comments:

1. Section 2. Results: in general, the results are adequately and extensively presented. The discussion, on the other hand, is rather limited and should also be expanded in relation to the current literature. Furthermore, authors should highlight the novelty of their work with respect to what has already been published.

2. Paragraph 2.1: The authors just show that the estimated content of TPA, TPC, and TFC of their extracts was significantly higher compared to previously published reports, but they do not discuss or speculate on the possible causes behind this difference.

3. Paragraph 2.3: The authors say that their extracts possess a “strong cytotoxic activity” against tumor cell lines. Actually, according to what is shown in figure 1, the cytotoxic activity can be considered relevant for very high concentrations (greater than or equal to 250 mg/ml for some lines or even 500 mg/ml for others) of the extracts. These high concentrations have little significance from a pharmacological or nutraceutical point of view.

Figure 1: I would suggest adding the name of the cell line being used at the top of each graph. This would make the reading and the correlation between text and picture easier and faster.

4. Paragraph 2.4: Again, the authors talk about “strong bactericidal activity”, while the activity is induced by very high concentrations of the extracts.

5. The English needs to be improved. The text contains several grammar errors that are noticeable at first sight. The authors should carry out some careful editing.

Author Response

We thank the reviewer for his/her time in reading the manuscript and providing helpful and insightful comments.

Comment 1: 

  1. Section 2. Results: in general, the results are adequately and extensively presented. The discussion, on the other hand, is rather limited and should also be expanded in relation to the current literature. Furthermore, authors should highlight the novelty of their work with respect to what has already been published.

It has been corrected.

Comment 2: 

  1. Paragraph 2.1: The authors just show that the estimated content of TPA, TPC, and TFC of their extracts was significantly higher compared to previously published reports, but they do not discuss or speculate on the possible causes behind this difference

It has been corrected. This was mainly due to the fact that the obtained preparation was purified from ballast substances.

Comment 3: 

  1. Paragraph 2.3: The authors say that their extracts possess a “strong cytotoxic activity” against tumor cell lines. Actually, according to what is shown in figure 1, the cytotoxic activity can be considered relevant for very high concentrations (greater than or equal to 250 mg/ml for some lines or even 500 mg/ml for others) of the extracts. These high concentrations have little significance from a pharmacological or nutraceutical point of view.

Figure 1: I would suggest adding the name of the cell line being used at the top of each graph. This would make the reading and the correlation between text and picture easier and faster.

It has been corrected. The name of each cell line in the graphs has been added.

Comment 4: 

  1. Paragraph 2.4: Again, the authors talk about “strong bactericidal activity”, while the activity is induced by very high concentrations of the extracts.

We thank the reviewer for this comment. In our opinion, the observed bactericidal effects at higher concentrations of the extracts may be due to the limited bioavailability of the active ingredients to the bacterial cells tested (CFU/ml) by the spot-on-lawn method. On solid media, cells grow as clusters (colonies), which may limit the penetration of bioactive compounds deep into the bacterial colony, in contrast to free-floating cells of the same bacterial strains in liquid cultures. A similar effect was observed with aqueous and alcoholic extracts of Planktochlorella nurekis at 100 mg/mL and 10 mg/mL when using the spot-on-lawn method, while bactericidal effects in liquid culture were maintained at lower concentrations of extracts, i.e. at 1 mg/mL, 100 μg/mL and 10 μg/mL, respectively. Based on the reviewer's suggestion, we have added an appropriate explanation in the Results and Discussion section.

Potocki, L.; Oklejewicz, B.; Kuna, E.; Szpyrka, E.; Duda, M.; Zuczek, J. Application of Green Algal Planktochlorella nurekis Biomasses to Modulate Growth of Selected Microbial Species. Molecules 2021, 26, 4038. https://doi.org/10.3390/molecules26134038

Comment 5: 

  1. The English needs to be improved. The text contains several grammar errors that are noticeable at first sight. The authors should carry out some careful editing.

It has been corrected.

Round 2

Reviewer 3 Report

The authors answered all the comments. The manuscript may be accepted in the present form.